# Prognosis and Health Management (PHM) of Solid-State Batteries: Perspectives, Challenges, and Opportunities

**Hamed Sadegh Kouhestani** [1], **Xiaoping Yi** [2], **Guoqing Qi** [2], **Xunliang Liu** [2], **Ruimin Wang** [1], **Yang Gao** [3], **Xiao Yu** [4] **and Lin Liu** [1,*]

1   Department of Mechanical Engineering, University of Kansas, 3138 Learned Hall, 1530 W. 15th Street, Lawrence, KS 66045-4709, USA

2   School of Energy and Environmental Engineering, University of Science and Technology Beijing, Beijing 100083, China

3   School of Electromechanic Engineering, North Minzu University, 204th Wenchang North Street, Xixia District, Yinchuan 750030, China

4   School of Electrical and Information Engineering, North Minzu University, 204th Wenchang North Street, Xixia District, Yinchuan 750030, China

*   Correspondence: linliu@ku.edu

**Abstract:** Solid-state batteries (SSBs) have proven to have the potential to be a proper substitute for conventional lithium-ion batteries due to their promising features. In order for the SSBs to be market-ready, the prognostics and health management (PHM) of battery systems plays a critical role in achieving such a goal. PHM ensures the reliability and availability of batteries during their operational time with acceptable safety margin. In the past two decades, much of the focus has been directed towards the PHM of lithium-ion batteries, while little attention has been given to PHM of solid-state batteries. Hence, this report presents a holistic review of the recent advances and current trends in PHM techniques of solid-state batteries and the associated challenges. For this purpose, notable commonly employed physics-based, data-driven, and hybrid methods are discussed in this report. The goal of this study is to bridge the gap between liquid state and SSBs and present the crucial aspects of SSBs that should be considered in order to have an accurate PHM model. The primary focus is given to the ML-based data-driven methods and the requirements that are needed to be included in the models, including anode, cathode, and electrolyte materials.

**Keywords:** solid-state batteries; prognostics and health management; physics-based approach; data-driven approach





## 1. Introduction

Over the recent decade, the global energy market has seen a sharp increase in adopting lithium-ion batteries (LIBs) as a reliable source of fuel for electric vehicles (EV), electronic devices, and medical instruments. According to the statistics that were reported by Electric Drive Transportation Association (EDTA), the number of EV sales in the United States market has increased from 345 vehicles in 2010 to 601,600 in 2022, with a total of 1.8 million EVs over the twelve-year sales period [1]. With ever-increasing demand for energy storage devices that are lightweight, sustainable, with higher life-cycle, LIBs have emerged as a universal solution due to their lower weight, higher energy density, relatively low self-discharge rate, and longer life cycle [2]. In spite of the vast superiority of LIBs over traditional fossil fuels, they still suffer from reliability and safety issues. Reports of occasional exploding of LIBs in EVs, mobile phones, and energy storage systems due to their high flammability, have called for a safer approach for the further acceleration of EV deployment [3].

The safety issues and flammability of LIBs can originate from several factors that are highlighted by thermal runaway in the circuit. Thermal runaway is a natural phenomenan

that is caused by overcharging or the occurrence of internal short circuit during the charging phase in LIBs that can create major safety risks. Characterized by severe increase of temperature in the cell, thermal runaway enhances the internal pressure in the cell and can evaporate the liquid electrolyte quickly leading to an instantaneous release of energy in the cell and eventually fire or explosion [3,4]. This issue has been addressed in many studies over the recent years, focusing on examining the root cause of thermal runaway and how to mitigate this factor [5,6].

Moreover, the narrow operating temperature range of conventional LIBs, flammability of solvents, a higher demand for ionic conductivity as well as the energy density of energy storage systems pose a crucial challenge to bridging the gap between the materials research and industrial mass production.

To overcome the aforementioned challenges, solid-state batteries (SSBs), in which the organic liquid electrolyte is substituted by solid electrolytes (SEs), have shown to be a proper substitute for LIBs by replacing the organic electrolyte with solid materials [7]. SSBs are nonlinear dynamical electrochemical systems with complex internal mechanisms. In contrast to LIBs, SSBs significantly improve component safety, electrochemical stability, energy and power density, and the durability of battery packs [8]. However, in order to make battery packs market-ready, reliability, health state, and operational safety of the batteries should be evaluated and guaranteed. In this context, battery prognostics and health management (PHM) has emerged as a reliable engineering discipline that ensures the safety and availability of batteries [9]. Battery PHM refers to a multifaceted advanced set of techniques that ensures the integrity and functionality of the battery systems. In particular, it follows a systematic framework to accurately predict batteries' state of charge (SOC), state of health (SOH), and remaining useful life (RUL) by estimating the products performance given the current degree of deviation and degradation and suggesting an optimal health management strategy [10]. Due to the complex nature of electrochemical and mechanical behavior of SSBs, predicting the remaining lifetime and SOH of batteries become an extremely difficult task. However, it is essential to accurately estimate the battery status to ensure the functionality and timely maintenance of the batteries under various operating conditions [9].

The status of batteries can be monitored through prognostics and health monitoring frameworks. Health monitoring is assigned with the task of detecting the underlying degradations and preventing the potential faults, while prognostics is responsible for predicting how soon a product will progress toward failure [10]. PHM techniques are normally comprised of three primary components: condition monitoring, data acquisition, and health diagnosis. Condition monitoring is concerned with battery performance and deriving the crucial aspects of battery performance such as voltage, current, charge, and discharge capacity. Data acquisition is assigned with the task of obtaining the required performance indicators and health diagnosis is responsible for estimating the state of health of the batteries [11].

Over recent years, numerous studies have focused on the PHM of machineries. Several of the most recent comprehensive reviews of PHM of machine tools and different popular models can be found in references [12–14]. To date, many investigations have dedicated their focus to reliability assessment and health prediction review of LIBs using PHM methods, and no significant attention has been given to health management of SSBs. The internal mechanism of SSBs differ from that of LIBs by the virtue of solid-state electrolyte which alters the electrochemical and mechanical performance of the batteries. As such, in order to create a smooth transition path from LIBs to SSBs utilization in electronic devices and electric vehicles, a systematic review of the recent advances and progress in the field of prognostics and health management becomes increasingly important. Furthermore, in order to make the vast utilization of SSBs in the current and future market possible, ensuring the safety and functionality of the solid-state battery packs becomes an important task. Despite significant advances in reliability modeling of LIBs including model-based, physics-based, and data-driven methods, due to the variation of chemical reactions in the

SSBs cell level, it is imperative to review the existing PHM methods that are used for SSBs and make suggestions regarding the future perspective.

Thus, this work aims to present the recent advances in prognostics and health assessment of SSBs by providing a holistic review of the current state of art technology. In particular, the research works that have been carried out to create the transition path from liquid state batteries to SSBs, are discussed from several outstanding perspectives. To the best of author's knowledge, no research studies has been carried out yet that provides a detailed overview of the existing PHM modeling of solid-state batteries.

In this work, first the idea behind solid-state batteries is elaborated that makes them different from conventional LIBs. Additionally, the underlying system state indicators that are used to characterize the health of SSBs are introduced and discussed. These indicators are the primary components of PHM frameworks and are employed to differentiate the efficiency of battery packs during their operational time. Then, in the corresponding sections, major PHM techniques are discussed in addition to the recent contributions of research studies in this field. The organization of this work is presented as follows: Section 2 of this investigation describes the basic characteristics of SSBs and how it differs from conventional LIBs followed by providing the major health indicators that are used in PHM methods. In Section 3, notable model-based, data-driven, and hybrid approaches are elaborated in the field of SSBs followed by Chapter 4 in which we will present the conclusion, challenges, and perspective of SSBs.

## 2. Solid-State Batteries

During recent years, solid-state batteries (SSBs) have been widely used in a variety of applications due to their superior characteristics, which consists of an Li metal anode, solid electrolyte (SE), and composite cathodes such as lithium cobalt oxide cathode (LiCoO$_2$ or LTO). Compared to Li-ion batteries, SSBs exhibit higher energy density and stability, which have attracted a lot of interest in the vast adoption in electronic devices as an alternative power source. A schematic profile of the typical structures of SSB is shown in Figure 1. The figure shows that the liquid organic electrolyte is replaced with solid-state materials.

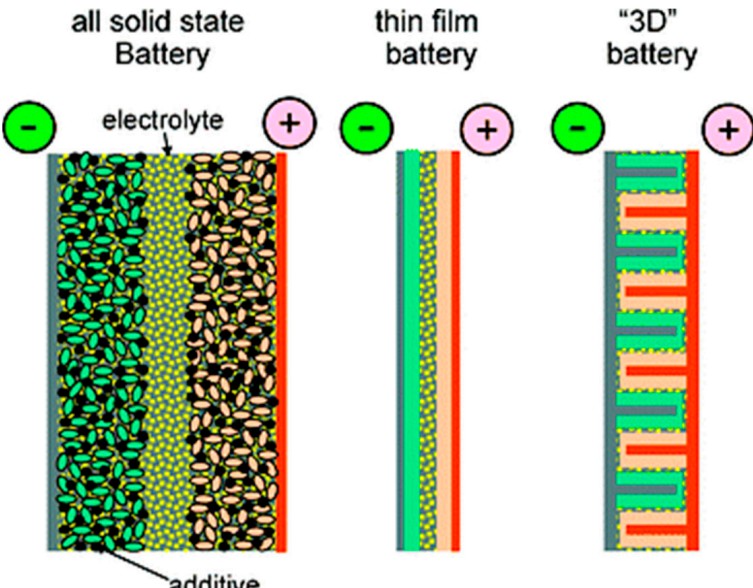

**Figure 1.** Schematic profile of typical structures of solid−state batteries. Adapted with permission from Ref. [15]. Copyright © 2022 Royal Society of Chemistry.

The working principle of SSBs is similar to that of LIBs. The typical chemical reaction process is as follows:

Cathodic reaction:

$$LiCoO_2 \xLeftrightarrow[Discharge]{Charge} Li_{1-x}CoO_2 + xLi^+ + xe^- \tag{1}$$

Anodic reaction:

$$C + xLi^+ + xe^- \xLeftrightarrow[Discharge]{Charge} Li_xC \tag{2}$$

Total reaction:

$$LiCoC_2 + C \xLeftrightarrow[Discharge]{Charge} Li_{1\times x}CoO_2 + Li_xC \tag{3}$$

Replacing the liquid electrolyte with a solid electrolyte not only enhances the battery's safety but also leads to a longer, more extended life and higher energy density. It is believed that SSBs with high safety, longer life, and high energy density will be a promising alternative to replace LIBs. In order to facilitate the large-scale application of SSBs and improve safety performance, it is necessary to develop a set of health management systems for SSBs.

The health status of batteries are assessed through several system state performance parameters, i.e., state of charge (SOC), state of health (SOH), and remaining useful life (RUL). SOC is concerned with estimating the remaining capacity during the operation of the battery before it is required to recharge it. SOH provides internal information regarding the health condition of the battery and its ability to deliver the nominal capacity with respect to the new battery. The RUL indicator is responsible for monitoring and predicting the failure of the battery from the degradation process. RUL is an important parameter that is used to measure the useful life that is left in the battery during its operational time.

The health characteristics of SSBs can be divided into two categories: external and internal characteristics. The external characteristics of SSBs refer to the quantities that were obtained by the simple processing of measurable data, mainly including Electrochemical Impedance Spectroscopy (EIS), discharge capacity, charging or discharging terminal voltage curve, and IC curve. The battery discharge capacity and internal resistance are the most direct external indicators of battery health. The battery charge-discharge capacity data itself can also be used as an external feature of the health state. For example, Hu et al. [16] used the terminal voltage data under dynamic charge and discharge conditions to obtain the sample entropy as a feature for battery capacity estimation. Feng et al. [17] obtained the probability density function through probability density statistics through experiments, and established a health state data table that was based on the function to estimate the health state of the battery online and in real-time. Zheng et al. [18] studied the discharge data, took Shannon entropy as an index, and combined it with the equivalent circuit model (ECM) to diagnose and locate the fault of the battery pack. Liu et al. [19] established the relationship between the features and battery capacity degradation by using external data features that were based on box Cox transform and support vector machine. The above-mentioned literature summarize the data acquisition methods for building external features and further elaborates on the impact of different features of the battery health management system. The acquisition method of external features is mainly used for battery health assessment and prediction based on empirical models or it is data-driven, which contains limited information. The aging models that were established based on this are mostly empirical models, which are vulnerable to data uncertainty and incompleteness, and have poor robustness and adaptability.

The internal health characteristics of the battery mainly refer to the internal physical and chemical parameters. The changes in these parameters characterize the degradation trend of the internal health state of the battery, or can be used as a tool to study the degradation mechanism. At present, many scholars have studied the changing trend of some parameters with aging, and used them as characteristic quantities to evaluate the health

state of batteries. For example, Han et al. [20] and Zhang et al. [21] identified the initial lithium intercalation rate and active material volume fraction of the electrode and used them as capacity attenuation factor analysis. Schmidt et al. [22] used the volume fraction of active substances as the characteristic of capacity attenuation, and the liquid phase conductivity as the characteristic of internal resistance rise. Ramadesigan et al. [23] identified the variation trend of solid-phase diffusion coefficient and electrochemical reaction constant with battery aging. Fu et al. [19,24] studied the effect of side reactions on battery degradation by using the degradation trend of active substance volume fraction, SEI film resistance, and electrolyte diffusion coefficient and established a physics-based degradation model.

## 2.1. Health Status Assessment of SSBs

The system state parameters that were discussed in the previous section can be further combined to form several important components to construct the PHM model of batteries. SOC is defined as the ratio of the remaining capacity of the battery to the maximum capacity and is described by:

$$SOC = \frac{C_R}{C_M} \times 100\%$$
(4)

where $C_R$ is the remaining capacity and $C_M$ is the maximum capacity.

It can also be calculated as:

$$SOC(T) = SOC(0) - \frac{\eta \int_0^T i\, dt}{C_n}$$
(5)

where $SOC(T)$ and $SOC(0)$ represent the SOC at time T and 0, respectively, $\eta$ is the Coulombic efficiency, $i$ is the current, and $C_n$ is the nominal capacity as a function of number of cycles n. The capacity of the battery is measured through the accurate monitoring of charging and discharging. This process can be somewhat challenging as the time that is required to capture such measurements can be time-consuming with high cumulative error.

SOH estimation is an essential component of the PHM that is characterized by the ratio of the maximum available capacity to the nominal capacity of the battery and is calculated by the equation:

$$\mathbf{SOH} = \frac{\mathbf{Q_{max}}}{\mathbf{Q_{nominal}}} \times 100\%$$
(6)

SOH measurement faces many difficulties due to the complex internal chemistry and operational condition of the battery. However, it is an important component of PHM that serves as a primary indicator for the time that the battery might need to be replaced. RUL deals with predicting the lifetime performance of the battery and estimate the threshold of failure that can be expressed as:

$$N_{RUL} = N_{EOL} - N_{ECL}$$
(7)

where $N_{RUL}$ is the RUL cycle number, $N_{EOL}$ is the end-of-life number, and $N_{ECL}$ is the equivalent circle life of the battery.

To give perspective into the utilization of health characteristics criteria in battery management strategies, many studies have been conducted to achieve such a goal. Zhang and Lee [25] reviewed the research progress on prognosis and health monitoring of LIBs, and summarized the algorithms for SOC prediction and remaining service life estimation. Watrin et al. [26] introduced three different adaptive systems (Kalman filter, artificial neural network, and fuzzy logic system), and analyzed their respective uses, advantages, and disadvantages. These three models can be used for SOC and SOH approximation. At the same time, Barr é et al. [10,27] summarized the development and related research of LIB health management systems in other fields, and briefly introduced the methods, algorithms, and models of battery RUL prediction and SOH approximation, as well as data-driven methods. Li et al. [28] reviewed the health assessment and health prognosis of LIBs based on data-driven methods. In 2019, Meng and Li et al. [10] conducted a literature review on

the prediction and health management (PHM) technology of LIBs. In 2020, Tian et al. [29] reviewed the SOH of LIBs, analyzed the causes of battery aging, summarized the SOH prediction methods, and further analyzed the main advantages and disadvantages of each technique. The above reports focus on the review of PHM and its related algorithms and mathematical models, further illustrating the important role of battery health characteristics criteria in battery management. The data acquisition process for RUL prediction of LIBs is shown in Figure 2, which can also be applied to SSBs. In this process, a specific battery cell is selected and reliable parameter data are obtained with the help of measuring elements. Then, these data are used to construct health indicators to estimate the RUL of batteries. The relationship between SOH and RUL is depicted in Figure 3. By estimating SOH and predicting RUL, the performance of the battery can be known in real-time and the battery life can be informed accordingly to ensure the safe and reliable operation of the power system or battery pack.

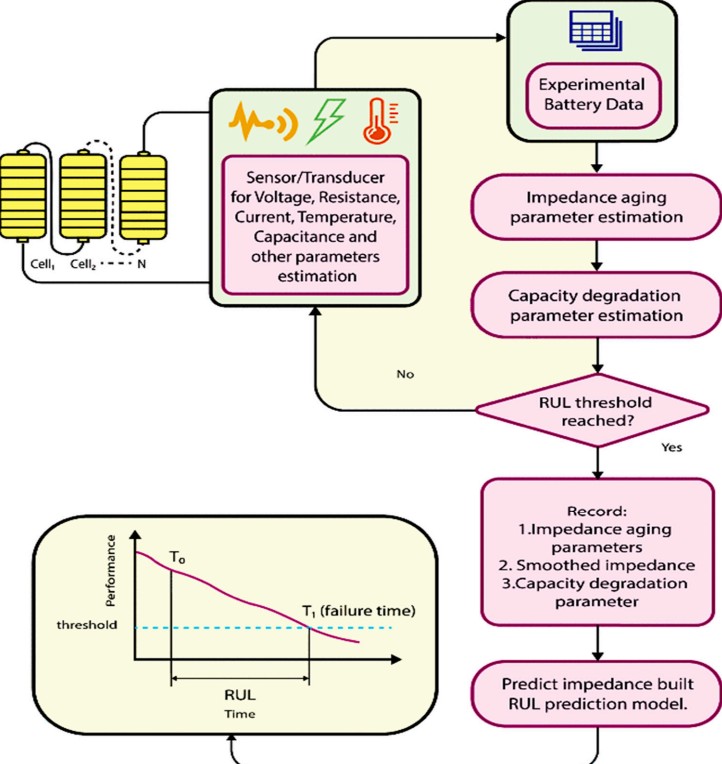

**Figure 2.** Data acquisition process for RUL prediction of LIBs. The blue line referring the assigned/designed battery performance or expectation; The blue line representing the actual battery performance. Adapted with permission from Ref. [30]. Copyright © 2022 Creative Commons Attribution 4.0 International License.

The SSB health management system is similar to liquid electrolyte batteries. Hence it is necessary to understand the health management strategy of liquid electrolyte batteries. The development of a literature review on the health management of batteries is discussed herein. Berecibar et al. [32] summarized battery SOH monitoring methods, reviewed the advantages and disadvantages of online BMS applications, and proposed a practical battery SOH estimation method. Cuma et al. [33] summarized the estimation strategies and methods that are used in EV. The above two papers summarize the research on SOH, but the recent literature is inadequate. Xiong et al. [34] discussed the classification of battery SOH estimation methods, elucidated the advantages and disadvantages of different methods, and proposed the future development prospects. Lipu et al. [35] briefly summarized the SOH and RUL estimation of electric vehicles, and studied the key problems and challenges of SOH and RUL. Hu et al. [36] systematically summarized the battery state estimation

methods, and briefly introduced the methods, problems, challenges, and development trends in this field. However, due to the large coverage of the literature, important details of the methods were neglected. Tian et al. [29] summarized SOH estimation methods, gave the definition and relationship of SOH and RUL, and put forward corresponding suggestions for existing problems. Similarly, Hu et al. [37] classified and summarized the RUL prediction methods for batteries. Ungurean et al. [38] conducted a review of the most relevant existing models, algorithms, and commercial devices that are used in embedded applications to estimate SOH/RUL. The working principle of this model was introduced and discussed in detail. Sarmah et al. [39] develop a hybrid method to accurately calculate the SOH of a battery in real-time and consider self-discharge, and then discussed the existing research results and future research directions. Ge et al. [31]. analyzed the development of SOH estimation and RUL prediction techniques for LIBs, summarized recent advances in direct measurement and model-based SOH estimation methods and data-driven and hybrid approach-based RUL prediction, and evaluated the advantages and disadvantages of each method. The SOH estimation and RUL prediction methods are shown in Figure 4.

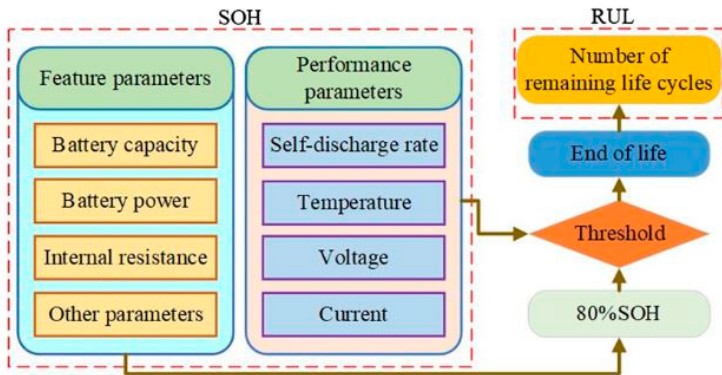

**Figure 3.** Relationship between SOH and RUL. Adapted with permission from Ref. [31]. Copyright © 2022 Elsevier Ltd.

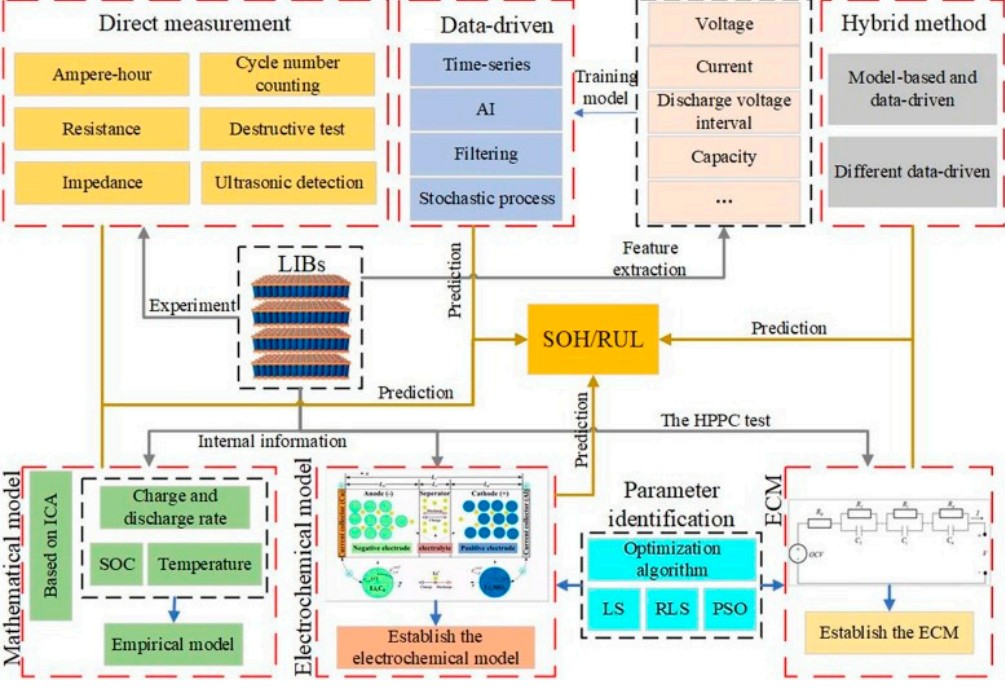

**Figure 4.** Methods for SOH estimation and RUL prediction. Adapted with permission from Ref. [31]. Copyright © 2022 Elsevier Ltd.

Yang et al. [40] reviewed the existing characteristic parameters of SOH in the cell-level and pack-level, and proposed several suggestions for the definition of SOH. A general prospect and estimation procedure for SOH is shown in Figure 5. The influence of external factors on battery degradation is introduced in this study, which lays the foundation for SOH estimation. In this report, the goals of SOH monitoring are discussed, and its applications are summarized from both short-term and long-term perspectives. Then, they discussed some key tasks and potential research directions from the following three aspects:

1. SOH characterization

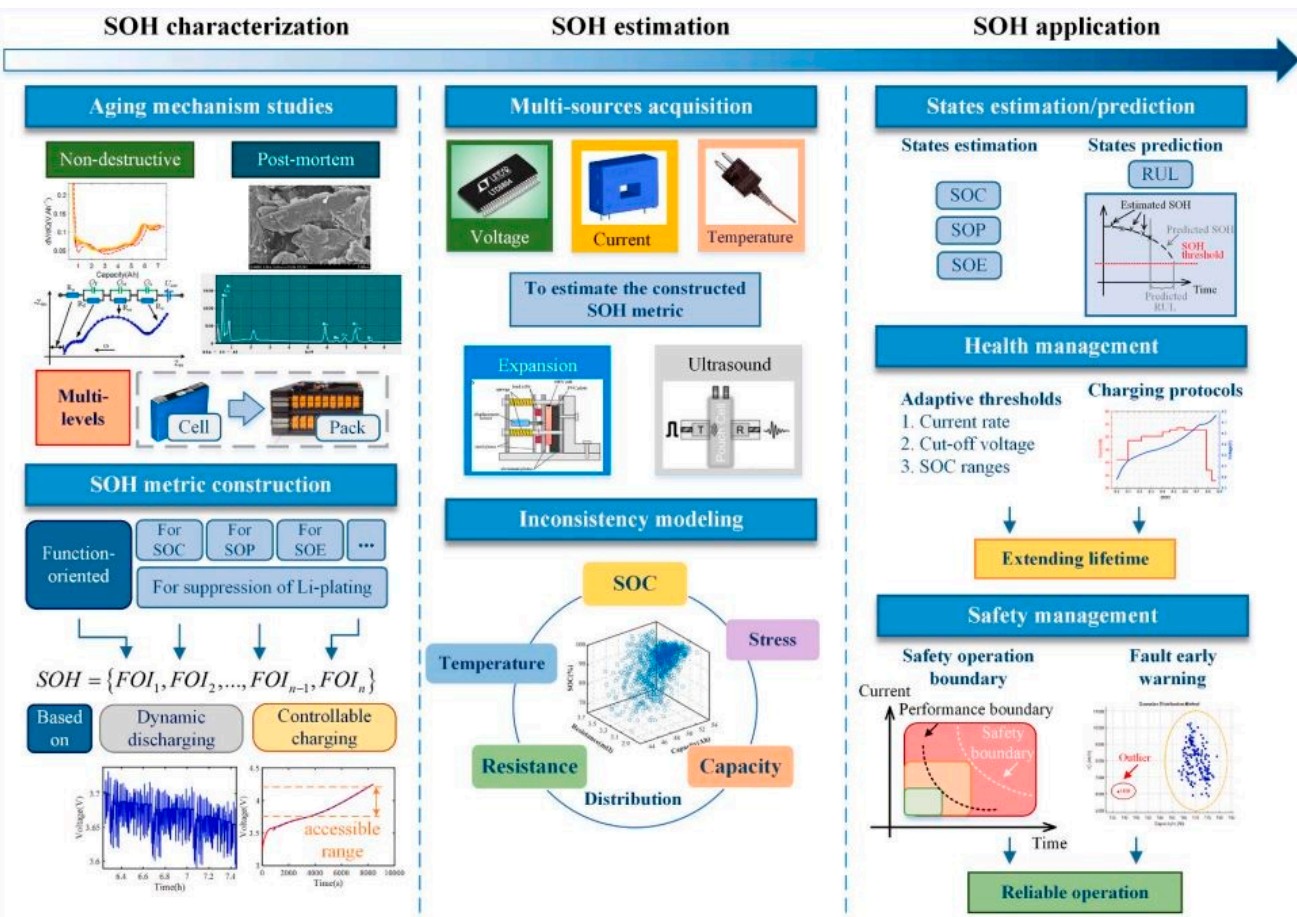

**Figure 5.** Prospects of SOH characterization, estimation, and application for batteries. Adapted with permission from Ref. [40]. Copyright © 2022 Elsevier Ltd.

2. SOH estimation

After developing a health management model, a time-domain analysis is needed, which plays a vital role in the improvement and optimization of the model. First, a trade-off needs to be made between model complexity and computational efficiency. Secondly, the extraction and estimation of SOH characteristics in the process of dynamic discharge and controllable charging is an essential task. In practice, there are very few instances where a battery can be completely exhausted or fully charged at the pack level. Therefore, the SOH estimation method should consider the working condition of the battery. Finally, the SOH estimation of fused multi-signals is likely to become a research hotspot in the future.

3. SOH application

Currently, there is still a large gap between SOH estimation and application. Current SOH prognostic methods, which mainly include short-term state estimation and long-term RUL prediction, are only pure judgments of battery retirement points, ignoring the significance of the battery aging process as a guide in health management. A prominent

outstanding issue is how to use the diagnostic results to extend battery life and ensure safe operation. The research on SOH indicators or parameters that are used to formulate battery health management strategies is still in its preliminary stage, which is a promising aspect of future research.

### 2.2. Battery Aging and Aging Characteristics

Generally, batteries are subject to irreversible processes (such as thermal and mechanical stress) and chemical changes (such as side reactions) during their operation, hence their performance will gradually degrade. The aging process involves many parameters and different degradation mechanisms. Generally, capacity loss information can be used as an indicator of battery aging. SOH is an indication of the end of an SSB's life and a measure of its condition relative to a new battery. Therefore, clarifying the aging mechanism of batteries plays an important role in the study of the SSB health management field. As shown in Figure 6, Hu et al. [41] summarized the battery failure process into two degradation modes: the loss of lithium inventory and the loss of active substances. Specifically, the loss of lithium mainly originates from the formation and decomposition of the solid electrolyte interface (SEI) membrane, electrolyte decomposition, and lithium electroplating [42]. The loss of active materials mainly stems from the electrical contact loss that is caused by graphite spalling, adhesive decomposition, collector corrosion, and electrode particle cracking [43].

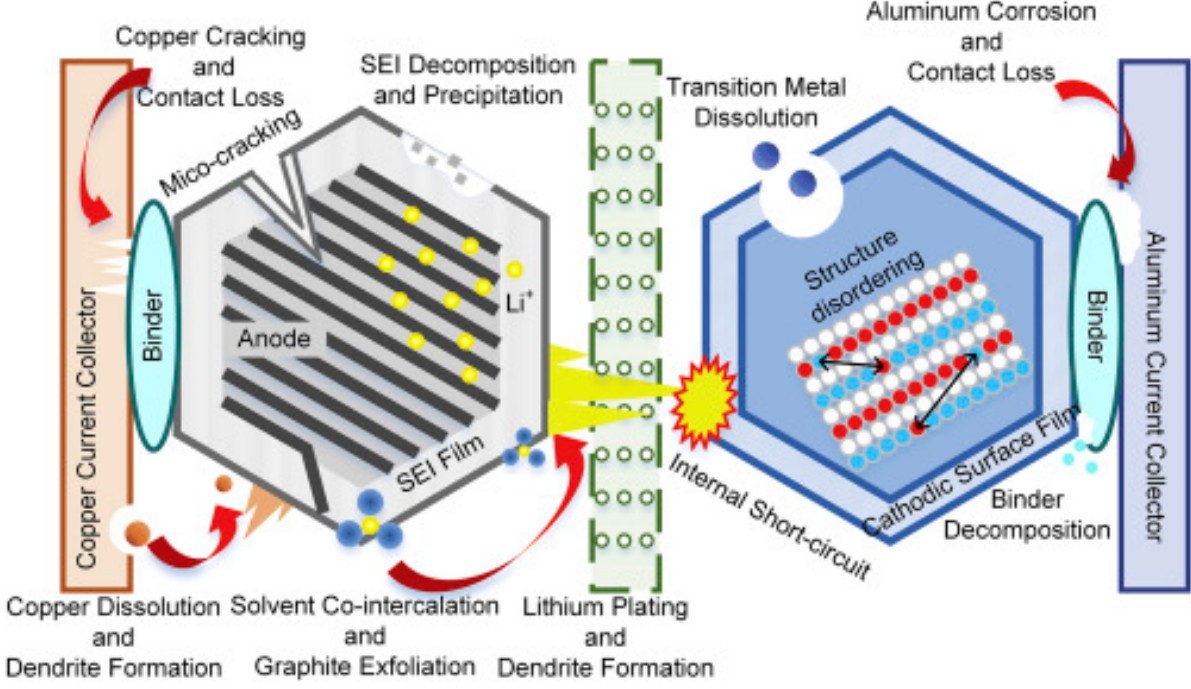

**Figure 6.** Main degradation mechanisms of LIBs. Adapted with permission from Ref. [41]. Copyright © 2022 Elsevier Ltd.

Tian et al. [29] studied the related investigations and attributed the battery aging to external environmental and internal factors. As shown in Figure 7, the external environmental factors refer to its working environment condition, such as temperature [44,45], charging and discharging rate [46,47], depth of discharge (DoD) [48,49], and charging cut-off voltage [46,50]. However, the internal factors mainly refer to three influencing mechanisms: lithium inventory loss (LLI) [51,52], active substance loss (LAM) [53–56], and conductivity loss (CL) [57,58]. LLI includes the formation of SEI layer, lithium dendrite, and battery self-discharge.

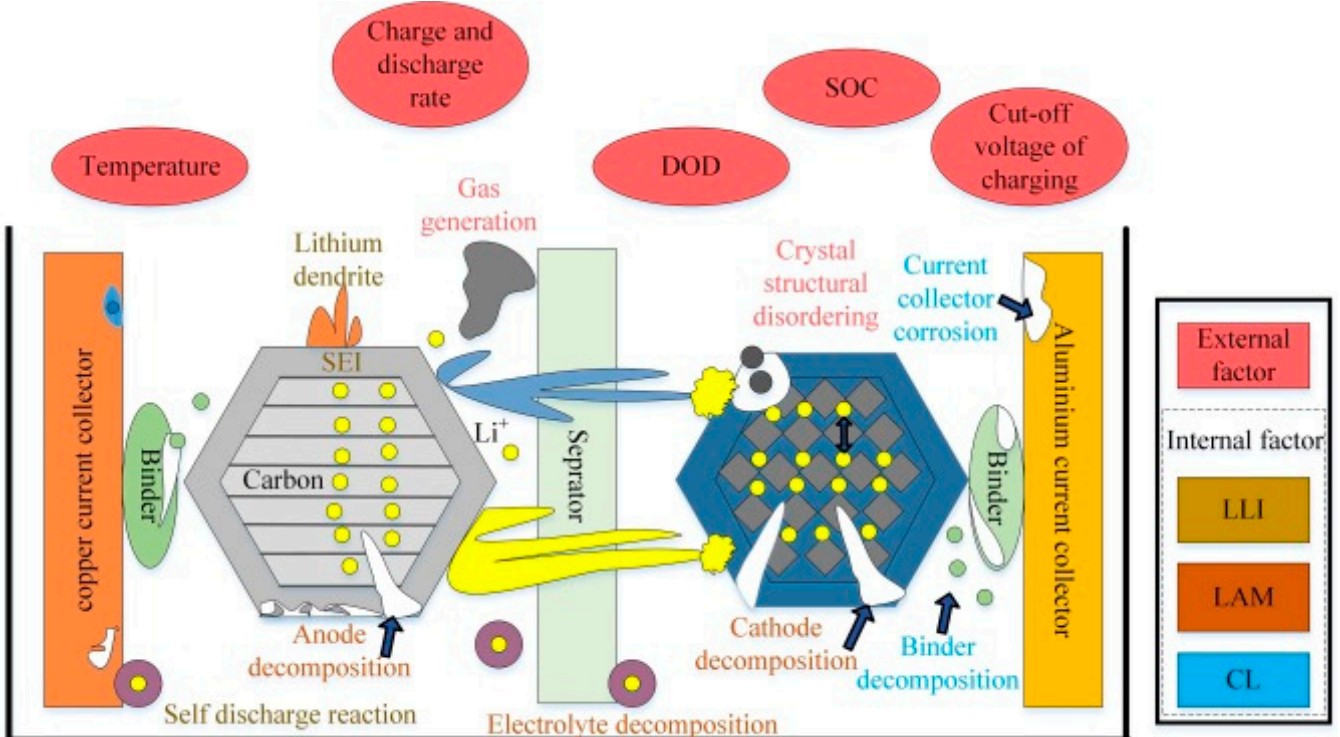

**Figure 7.** Schematic diagram of the causes of battery aging. Adapted with permission from Ref. [29]. Copyright © 2022 Elsevier Ltd.

## 3. Prognostics and Health Management (PHM)

PHM techniques devise a set of technologies and approaches to support condition monitoring, health status, fault diagnosis, and maintenance to maximize the reliability and safety of battery systems. They are mainly divided into three categories: model-based (physics-based), data-driven, and hybrid approaches. Each of these methods will be discussed in detail in the following sections.

### 3.1. Model-Based (Physics-Based) Approaches

The physics-based approach aims to establish a mathematical model that describes the degradation behavior of a battery accurately. It can be divided into four groups: electrochemical (mechanistic) models, ECMs, empirical models, or fused models.

#### 3.1.1. Electrochemical (Mechanistic) Models

The electrochemical model mainly completes the accurate modeling of the battery by describing the electrochemical reaction process inside the battery [59]. However, the differential equations that are established by this model usually contain a large number of unknown parameters, which increases the complexity of this model. In addition, the input variables in the electrochemical model include a large number of internal battery parameters, such as the effective area of the electrode space, ionic conductivity of the electrolyte, average distance from the electrode to the current collector, etc. These parameters are normally difficult to obtain by measuring the external structure and cycle performance of the battery.

The electrochemical model is established based on electrochemical principles, which can reflect the relationship between the external characteristics and the internal parameters of the battery and can describe the change in the output characteristics during the charging and discharging process. The types of SE (LiPON [29,42,60], $Li_3PO_4$ [61,62], LiTFSI [63], LATP [64], $Li_6PS_5Cl$ [65], and polymers [66]), anode (lithium metal [42,60,61,67,68] and

graphene [69]), and cathode (LiCoO$_2$ [42,60,68] and TiS$_2$ [60,67]) in the macroscopic model can be adjusted according to the actual application. The electrochemical model can be used to analyze the charge-discharge behavior and cycle performance of the battery. In addition, the effects of contact areas and compressive pressure [63], the contributions of individual overpotential and impedance, the definition of diffusion coefficient, the expansion of the electrode particles that is caused by intercalation [70] can also be evaluated.

Danilov et al. [61] proposed a mathematical model for SSBs that takes into account the insertion and de-insertion of Li$^+$ at the electrolyte/electrode interface and ignores the side reactions that are occurring at the interface. In this study, the discharge curves at different current densities are in good agreement with the experimental results [61]. Ansah et al. [65] also studied the effect of structural parameters on the discharge performance and found out that increasing the thickness of the cathode and decreasing the thickness of the electrolyte were beneficial to increasing the battery capacity. However, due to the complexity of SSBs, it is difficult to validate the results. Considering that several parameters, such as the mobility number and diffusion coefficient of Li$^+$, are not suitable for SSBs, Fabre et al. [42] then employed various electrochemical techniques such as galvanostatic intermittent titration technique (GITT) and electrochemical impedance spectroscopy (EIS) to modify these parameters. Kodama et al. [71] used nonlinear stress analysis to calculate the ionic conductivity and showed that the simulation results between the diffusion coefficient and the concentration of Li$^+$ are consistent with the experimental results [72]., Liu et al. [68] proposed an improved Planck–Nernst–Poisson and Frumkin–Butler–Volmer (MPNP-FBV) electrochemical model and considered the influence of electric double layer (EDL) structure and vacancies to investigate the essential phenomena at the equilibrium state in SSBs. They found that the total electrostatic potential drop at equilibrium is related to the difference in free enthalpy between different materials. In addition, the charge transfer resistance of the diffused bilayer structure is higher than that of the dense bilayer structure.

Actually, SEs are fragile that even slight volume changes in SSBs could cause particle fracture, disconnection, and eventually pulverization [21]. To improve the interface contact quality and reduce the interface impedance, several methods have been proposed, including depositing buffer layers between SEs and the electrodes by pulsed laser deposition (PLD) or atomic layer deposition (ALD) [21,73,74], grinding nanocomposites to reduce the particle size and increase surface area [75,76], and fabricating composite cathodes [70]. Tian et al. [60] and Shao et al. [60] introduced a parameter to describe the contact area, which adjusts the current density in the 1-D Newman model. They found that the capacity drop was correlated with the loss of contact area, and the optimal charging performance could be obtained under medium compressive pressures (0.4–1 MPa). To illustrate the relationship between lithiation-induced stress evolution and electrode structure, Fathiannasab et al. [64] presented a 3D model by using a synchrotron transmission X-ray microscopy tomography system to reconstruct the morphology of the SSBs. They revealed that SE with lower stiffness can decrease stress in the microstructure, but aggravate the anisotropic displacement of AM particles. Interestingly, the anisotropic displacement of AM particles can also be prevented by applying external compressive pressure. In addition, bending also has a significant effect on battery performance, as bending the SSB from anode to cathode when a force is applied can reduce the cell potential, while bending in the opposite direction induces a potential change and leads to a reduction in lithiation ability [77].

Becker-Steinberger et al. [78] proposed an SSB model that takes into account ion transport in crystalline metal oxide solid solutions. Specifically, the diffusion part of the electric double layer is dynamically described by the Poisson equation, while the Stern layer potential drop is modeled by the Robin boundary condition. In addition, electrochemical reactions at the electrode/electrolyte interface (EEI) are modeled with nonlinear Neumann boundary conditions. Danilov et al. [61] developed an isothermal SSB model that takes into account the incomplete dissociation of ions in the electrolyte. The model consists of two partial differential equations (PDEs) describing the diffusion process in the SE and cathode. Based on this model, Kim et al. [62] ] proposed battery management algorithms such as

state estimation. However, the equation order of this method is higher, and the efficiency of the algorithm has not been determined. Deng et al. [79] used a combination of Padé approximation, polynomial profile approximation, and equal response coefficient assumptions to reduce the rigorous PDE model with significantly reduced computational burden and high fidelity that is suitable for online parameter estimation and condition monitoring. Despite the high accuracy of electrochemical models, these models have shortcomings such as complex model structure, difficult parameter identification, and low computational speed. In order to improve the robustness of the models, alternative approaches have been adopted with higher accuracy, such as P2D (Pseudo Two-Dimensional) model, single particle model, and the electrode average model. In addition, there are also methods such as polynomial approximation, Padé approximation, finite volume discretization, and orthogonal decomposition, which are less prevalent.

### 3.1.2. Equivalent Circuit Models

Even though electrochemical models offer efficiency for battery modeling and system state predictions, due to the difficulty in solving the coupled partial differential equations and the high demands on model parameterization and computation time, it has been attempted to propose alternative models that can obviate these challenges. SSB is highly nonlinear and time-varying during the operational time, and internal parameters such as internal resistance, SOC, SOH, and self-discharge parameters during battery operation cannot be obtained by direct measurements. In order to analyze the relationship between the internal variation law of the battery and related parameters, establishing an equivalent model of the battery is an effective research method [80]. To achieve the expected high performance, practical applications of SSBs require accurate and computationally efficient models onboard management algorithms so that the SSB safety, health, and cycling performance can be optimized under a wide range of operating conditions. ECM has been widely used in battery management modeling as a compromise between accuracy and feasibility. In the ECM, the power batteries' dynamic response, static characteristics, and dynamic polarization effect are described by ideal electrical components, constant voltage source, and RC network, respectively. This strategy has been successfully applied to the estimation of SOC, SOH, and SOE due to its simple model equations, convenient parameter identification, and good real-time performance. ECM is a semi-empirical model, which describes the charge and discharge characteristics of the battery by arranging and combining electrical components such as voltage sources, resistors, inductors, and capacitors [81].

The system parameters of the electrical components in ECM models can be determined by combining different parameter identification methods, such as extended Kalman filtering, particle filtering, and other algorithms. At present, the development of ECMs is relatively complete, and the more frequently used models include Rint, first-order RC, second-order RC, and PNGV [82]. In theory, the multi-order models have higher accuracy than low-order models, but they have less advantages in accuracy and computational efficiency due to the large number of parameters that need to be identified and the inevitable errors in each parameter.

In the Rint model, the battery is treated as the series connections of the ohmic internal resistance and the ideal voltage source, and polarization effects are not considered. The Rint model is characterized by a simple structure and the least number of parameters. Notably, the error of the model increases with the increase of the charge and discharge rate of the battery [83]. The first-order RC model, also known as the Thevenin model, is composed of an ideal voltage source, an ohmic resistance, a polarization resistance, and a polarization capacitor in series. Compared with the Rint model, the first-order RC model describes the polarization effect of the battery during charging and discharging. In addition, the first-order RC model has high accuracy in terms of constant temperature and constant current charging and discharging conditions, and can realize the estimation of the state parameters of LIBs. However, the aging or temperature change of the LIB will cause the

internal impedance characteristics of the battery to change from a single impedance arc to a double impedance arc, which significantly impacts the accuracy of the battery model [84].

Compared with the first-order RC model, the second-order RC model adds a series-connected RC network to describe the battery's electrochemical polarization and concentration polarization at different stages of the charging and discharging process. The second-order RC model is more computationally expensive than the electrochemical model and the ECM. Notably, the second-order RC model has high accuracy in describing the dynamic polarization behavior of LIBs under high-rate current, and the operating results are closer to the actual state of the battery, and therefore, are widely used in the research of single batteries [85].

The PNGV model is a derivative model of the first-order RC model, which can describe the battery capacity while reflecting the DC response characteristics. The primary working principle of PNGV models is that a capacitor is connected in series based on the first-order RC mode. The purpose of this model is to describe the relationship between the open circuit voltage of the battery and the accumulative charge and discharge capacity. Then, the model can realize the estimation of the available capacity of the battery, and is mostly used to assess the SOH [82]. The general nonlinear (GNL) model is also called the nonlinear equivalent model, which is derived from the generalization and development of the Rint model, Thevenin model, and the PNGV model. The addition of circuit components in the GNL model makes the physical meaning of each part clearer, so that the voltage change process can be better simulated.

Since the parameters in the ECM are closely related to the working state of the battery, the effectiveness of parameter identification during battery operation is crucial. The current identification methods mainly include the nonlinear least squares method, neural network algorithm, and bionic optimization algorithm [86–88] Among them, bionic optimization algorithms include genetic algorithm, particle swarm algorithm, simulated annealing algorithm, and so on [89]. The ECM method is one of the most commonly used single-cell models in modeling LIB and SSB packs. The output characteristics and accuracy of the battery pack model depend on the series-parallel sequence and modeling method of the selected single-cell model. However, the current production level and manufacturing process are difficult to ensure the consistency between individual cells. Therefore, the single-cell model cannot accurately represent the predictive model of the battery pack through simple quantitative accumulation. The ECM models generally focus on the external physical quantities such as terminal voltage and current. Moreover, ECM models do not reflect the electrochemical properties and complex variations in the microstructure of the battery. The main modeling principles of the ECM method for a single cell are: (1) The model includes the chemical reaction mechanism of the battery, and the relevant model parameters should be identified; (2) the model can accurately reflect the characteristics of the battery and adapt to different environments and working conditions; (3) the single-cell model is as simple as possible within the scope of the design requirements to simplify the calculation process and improve the usability of the model.

The physical structure of SSBs is fundamentally different from conventional liquid electrolyte-based Li-ion batteries. A suitable SSB model with high fidelity and a low computational burden is essential for most model-based management algorithms. The ECM has the advantage of being easy to implement in a wide range of applications, in particular, in circuit simulation and control system design software packages such as MATLAB/Simulink. In these packages, various numerical solvers have been included, which can be selected to solve circuit models and facilitate control system design. However, ECMs lack mechanistic insight into electrochemical dynamics, have limited applicability for battery performance prediction under wider operating ranges and changing system dynamics, and fail to properly address battery degradation and internal safety issues. Additionally, modern applications of batteries need to be designed for increased load dynamics, higher current rates, and harsher operating environments. In this case, the

functional complexity, model order, and testing effort for identifying ECM parameters must increase substantially to achieve sufficient extrapolation.

### 3.1.3. Empirical Models

Building mathematical models is an effective method to analyze and optimize the performance of batteries. Generally, mathematical models are divided into two types: one is a mechanism model that is established by theoretical analysis; the other is an empirical model, which is proposed on the basis of experiments and can simulate the performance and explain the behavior of the battery to a certain extent. The former is often used to describe the mechanism by which various factors affect battery performance, while the latter is often applied to simple simulations of the performance of single batteries in a battery pack. In addition, there are also semi-empirical models that combine mechanism and empirical models. The empirical model refers to the model description of the cell capacity decay process that can directly or indirectly reflect the change law of the state variable of the capacity decay with time, the total discharge capacity or the number of cycles through the empirical formula. In the process of estimating the SOH of lithium batteries, the empirical model usually needs to use statistics to process the data to determine the initial parameter values of the model.

The modeling and simulation of LIBs has always been a research hotspot in the field of electrochemistry. However, most existing reports have studied the numerical and empirical models of LIBs separately. Establishing an LIB degradation model is one of the important links in predicting cycle life. However, it is difficult to establish an accurate battery degradation process model that is based on the electrochemical processes inside LIBs that are under actual operating conditions. In addition, the degradation process of the battery is directly affected by various factors such as temperature, impedance, end of charged voltage (EOCV), and depth of discharge (DOD). Generally, the more parameters that are involved in the model, the higher the accuracy of the model. However, some parameters are not easy to obtain in the process of battery capacity degradation, and the model establishment is more complicated. Empirical models that are commonly used in LIBs include internal resistance model, ECM, neural network model, fuzzy algorithm model, and the genetic algorithm model. Saha et al. [90] first carried out experiments on the performance degradation of LIBs under different conditions and obtained a large amount of test data. Some scholars conducted accelerated life tests on lithium power batteries at multiple temperatures (40–70 °C) [91,92]. They proposed a completely empirical model according to the variation law of the battery's internal resistance, temperature, and SOC, and finally developed a multi-sigmoid model. Ramadass et al. [93] quantitatively studied the capacity fading of batteries through the changes of SOC, resistance, and diffusion coefficient of an SEI membrane, and proposed a semi-empirical model for battery capacity degradation. Furthermore, Ning et al. [94] improved a semi-empirical model that was based on the quantitative analysis of the effects of EOCV and DOD on the cycle life of batteries.

### 3.1.4. Fused Models

Fused models combine all available knowledge, information, and data sources, bringing the advantages of model-based and data-driven approaches. Specifically, fused models can combine the robustness and interpretability of model-based methods with the specificity and accuracy of data-driven methods. They incorporate different types of battery models to extract additional features from the available data. The differential equations that are established by the electrochemical models usually contain a large number of unknown parameters, and the input variables include many internal parameters of lithium batteries, which are difficult to obtain by measuring the external characteristics of lithium batteries. Therefore, electrochemical models are rarely used in practical BMS. The ECM avoids the extensive use of internal parameters of lithium batteries and reduces the difficulty of establishing the model. In order to achieve high-precision modeling, Verbrugge et al. [95,96] introduced a first-order delay in the RC model of lithium batteries, which was experi-

mentally demonstrated to have better performance under dynamic current conditions. Plett et al. [83] obtained the fused model by fusing the Shepherd model, the Unnewehr general model, and the Nernst model. Liu et al. [97] modeled the lithium battery that was based on the fusion model and achieved an accurate estimation of the state of the lithium battery based on the improved fusion model.

The RUL is usually defined as the number of remaining charge and discharge cycles when the battery reaches the end of life (EOL), which is used to measure the reliability of the battery in its life span and is a description of the future state of the battery at the macro scale. However, SOH belongs to the description of the current state of the battery at the macro scale. To fully evaluate the aging degree of the battery, it is necessary to perform SOH estimation and RUL prediction simultaneously [35]. So far, joint estimations of SOC-SOH are now common, but predictions of RUL are usually done separately. Since it is difficult to achieve an accurate prediction of RUL using only a single method, fusion algorithms are regarded as the main research direction. In addition, RUL is a description of the future state of the battery, and therefore, it is necessary to provide an uncertainty expression of the predicted results to improve reliability. Xing et al. [98] proposed a fused model method that was based on exponential and polynomial to achieve RUL prediction by using PF to update the model parameters online. These types of models are simple, but usually only provide point forecasts and perform poorly in long-term RUL forecasts. In addition, data-driven methods using machine learning are also widely used in RUL prediction. Wang et al. [99] established a multi-step capacity prediction model, which takes energy efficiency and average operating temperature as the input of SVM, and the current capacity of the battery and the decreasing value of the capacity during the cycle as the output of SVM. Although machine learning algorithms can achieve accurate modeling of nonlinear systems, they have poor multi-step iterative prediction ability, and usually only single-step prediction can be performed [100].

### 3.1.5. Comparison of Physics-Based Approaches

Although the electrochemical model has high accuracy, it has obvious shortcomings such as complex model structure, difficult parameter identification, and low operation speed. Therefore, the electrochemical model is not suitable for the BMS of the actual vehicles. The electrochemical model mainly realizes the accurate modeling of the lithium battery by describing the electrochemical reaction process inside the battery. However, the differential equations that are established by the electrochemical model usually contain many unknown parameters, which increases the complexity of the electrochemical model. Meantime, the input variables in the electrochemical model include many internal parameters of the lithium battery, which are difficult to obtain by measuring the external characteristics of the lithium battery, such as the effective area of the electrode space, the ionic conductivity of the electrolyte, the average distance of holes from the electrode to the current collector, etc. [101]. The ECM simulates the lithium battery by establishing a circuit, which avoids the extensive use of internal parameters of the lithium battery and reduces the difficulty of model establishment. Therefore, ECMs are frequently used in practical BMS test systems. Empirical models are easier to obtain and more applicable. Furthermore, the prediction of the remaining life of the lithium battery can be realized by using an appropriate filtering algorithm combined with the corresponding empirical model. However, the empirical models are difficult to describe the influence mechanism of multiple factors on the aging and load dynamic characteristics of lithium batteries, and the accuracy and stability of the model still have certain limitations. Based on the fusion model, higher-precision modeling of lithium batteries can be achieved. Importantly, physical models and simulation technologies keep pace with the time and develop vigorously. The development of new calculation methods to accelerate the understanding of SSBs is particularly important due to challenges such as the complexity of the interface and the diversity of SEs. The combination of powerful simulation techniques is currently a topic of great interest, especially utilizing machine learning techniques. The market demand for

batteries with excellent performance will drive the innovation of multi-functional physical models and high-efficiency computational methods in the future, which is essential to break through the bottleneck of the development of battery technology. Due to the advantages and limitations of each method, there is no single perfect method for battery PHM. As far as physics-based forecasting methods are concerned, they can use limited data. As battery systems, operating conditions, and monitoring data become more complex, a data-driven approach that is based on machine learning becomes increasingly beneficial. Hybrid approaches are also promising in the field of battery PHM by integrating data-driven and physics-based approaches.

*3.2. Data-Driven Approaches*

Despite significant efficiency of model-based approaches, in reality, due to large the volume of dimensionality and complexity of electro-chemical and mechanical characteristics of batteries, adopting data-driven methods can offer substantial improvement over model-based approaches. Data-driven techniques can transform high dimensional and noisy data into lower dimensional and cleaner ones that can be used for prognostications. They employ a set of predictive models that are dependent upon the data's quality and size. Using historical monitoring data and statistical pattern recognition tools to detect faults, data-driven methods predict the degradation of a battery based on a training database of internal and external covariates. This method's three major tasks are fault diagnostics, prognostics, and condition-based maintenance. Here, a handful of the most known data-driven approaches that can be used for SSBs are presented and discussed. This section aims to compile the most prevalent and robust data-driven methods that can serve as a benchmark for clients to select the appropriate method for SSB health status assessment.

Data-driven approaches that are used in PHM generally can be classified into two categories: statistical approach and machine learning (ML) approach. Statistical-based data-driven methods generally rely on statistical parameters of the dataset such as the standard deviation, covariance and mean, that are contingent upon the existence of probability distribution of the statistical parameters [102]. On the contrary, machine learning methods do not require any statistical assumption and makes a prediction that is based on the acquired data. Broadly speaking, ML methods are widely recognized as the primary data-driven approaches. They can be divided into three categories: supervised learning, unsupervised learning, and semi-supervised learning. The main difference between the three is the type and amount of data that are available. Most ML-based data-driven prognostication models are built using supervised learning models. Supervised learning can predict the output values of continuous quantities (such as volume modulus, band gap, etc.) or discrete quantities (such as crystal structure, etc.). Unsupervised learning models are often used to classify or reduce the dimension of vectors, which solves the problem of creating from sparse datasets. Semi-supervised learning is rarely used in SSE prediction. The basic rule of this method is to use some local features of labeled data and the overall distribution of the unlabeled data to obtain acceptable classification results.

Among many ML techniques that offer significant capability for battery informatics, the most important underlying feature is the capability of the method to accurately include the inherent properties of the battery material and behavior. It starts with predicting the performance of the material for a targeted functionality that normally uses parametrization in one or more crucial material properties. Then, the established model is used to predict the functionality of the material with the best given performance. An important feature of the data-driven method that is employed for solid-state electrolytes, which is the governing factor that differentiates them with liquid LIBs, is the ability to accurately represent the compositional information and crystalline structure of the electrolyte in the model. Deep learning data-driven methods have achieved substantial breakthroughs in representing these features in their model [37,103]. They possess the potential to transfer information from the learning process of the formation energy in order to represent elemental knowledge. Furthermore, the recently developed crystal graph convolutional neural network

(CGCNN) has illustrated the ability to accurately represent the crystalline structure [104]. In the pool of various data-driven algorithms for establishing SSB models, the most notable methods are artificial neural network (ANN) [105], kernel ridge regression (KRR) [106], support vector machine (SVM) [107], k- Nearest Neighbor (kNN) [108], Random forest (RF) [109], and the Bayesian method (BM) [110].

### 3.2.1. Artificial Neural Network (ANN)

Artificial neural network (ANN) motivated from the biological arrangement and characteristics of the human brain is a widely used method in data-driven approaches that is based on a collection of connected neurons. Each connection transmits or receives a signal from adjoining neurons that connected to them. The model is constituted by three components: the input layer, hidden layer and the output layer and are classified into different types where the primary used methods are: feedforward neural network (FNN) and recurrent neural network (RNN). Their difference originates from the fact that in RNN, the connections between the nodes form a directed or undirected graph, while in FNN, the connections do not form a cycle. Many researchers over the recent decades have focused on employing the ANN methods to perform health diagnosis and SOC estimation of LIBs and SSBs [105].

### 3.2.2. Kernel Ridge Regression (KRR)

A generalized version of linear regression and ridge regression methods, KRR extends the linear regression into a nonlinear correlation between the available data and maps them into a higher-dimensional feature space. The nonlinear regression scheme is then transformed into a linear format in the feature space [110]. Due to the difficulty in selecting an appropriate mapping function, the kernels are applied, representing a similarity between the inputs. For this purpose, a non-linear kernel function is applied in the input space instead of mapping the data and solving high-dimensional non-linear regression. Examples of kernel methods are Gaussian, polynomial, and Laplacian kernel. Fitting the KRR models are commonly a challenging task due to computationally intensive demand of the data which limits the application of medium-sized datasets [106].

### 3.2.3. Data-Driven Prognosis (DDP)

Recently, a novel data-driven method called data-driven prognosis (DDP) was proposed by Chandra et al. [111] that relies on in situ data measurements and estimates the system's failure based on the curvature information that is extracted from the system. This method was later on employed to analyze LIBs [112]. The proposed approach extracts the constitutive parameters of LIBs in the shape of curvature and analyzes the curvature information in the system based on the pairwise information of the data points. Then, it estimates the probable timeframe that the system might enter the instability stage and using a set of threshold criteria, predicts the failure of the system [112].

### 3.2.4. Support Vector Machine (SVM)

SVM methods align with KRR techniques in that they attempt to solve linear classification problems in a high-dimensional feature space [113]. The principle the SVM function under is to locate a hyperplane in the feature space and classifies the data points, and identify the plane with a maximum distance between the data points [107].

### 3.2.5. k-Nearest Neighbor (kNN)

A commonly employed nonlinear ML method, kNN is used to solve both classification and regression problems. It hypothesizes that similar data points are in the vicinity of feature space and classifies the new data points into a category governed by its kNNs [108]. This procedure is counted as the classification task. In order to perform, regression, among the kNNs, a weighted average label value is calculated. a requirement for a reasonable distance metric is a limitation of this method [108].

### 3.2.6. Random Forest (RF)

RF is an ML technique that is used for classification and regression problems [109]. It constructs an ensemble of decision trees on various domains of the data points, makes a prediction that is based on each decision tree, and calculates their mean. Higher accuracy and less overfitting is achieved by selecting a large number of decision tress. The employment of ensemble-based architecture in RF methods leads to high accurate predictions and efficiency in the results [109].

### 3.2.7. Bayesian Method (BM)

BMs are an optimization approach that are used to produce a probabilistic model for a target function commonly utilizing Gaussian Processes (GPs) [83,86]. GPs use stochastic procedures to describe probability distributions over the functions and assigns a probability to each of these functions. Then, the probability distribution is used to represent the most probable characterization of the data. A notable advantage of Gaussian process regression is their ability to describe the uncertainty of each prediction model. On the other hand, it might be computationally intensive and time-consuming [87].

### 3.3. Application of ML-Based Data-Driven Techniques in Solid-State Batteries Research

In order to accurately model the SSB characteristics in the selected data-driven framework, the SSB primary components, i.e., the electrodes (anode and cathode) in addition to electrolytes, should be modeled properly. The candidate SSB cathode materials should possess high energy density, high voltage and capacity, and stable mechanical properties. Multiple data-driven methods have been utilized to model the materials in a way that they match these requirements.

### 3.3.1. Anode Materials

Eremin et al. [88] integrated topological analysis with density-functional theory (DFT) modeling in addition to ridge regression in the configurational space of $LiNiO_2$ and $LiNi_{0.8}Co_{0.15}Al_{0.05}O_2$ cathode materials. They showed that the topology of Li layers, ions, and dopants substantially influence the energy balance. In a similar study, Natarajan et al. [89] combined ANNs with adapted cluster functions to predict the formation of Li-vacancy orderings on the spinel $LiTiS_s$ and demonstrated that the ANN method can produce the DFT-computed convex hull with information regarding the pair cluster correlation as the input variable. In 2020, Eckhoff et al. [114,115] used an ANN method to model $Li_xMn_2O_4$ that utilized a Jahn–Teller distortion model to predict several properties of SSBs such as an Li diffusion barrier and phonon frequencies and oxidation. Bartel et al. [116] investigated seven ML methods to study the formation energy of Li transition metal oxides using chemical formula and showed that models can predict the formation energies accurately.

### 3.3.2. Cathode Materials

Similarly, several studies have been carried out to simulate the SSBs anode materials using ML-based data-driven methods. Artrith et al. [117] employed atomistic ANN models to evaluate the crystal structure of $TiO_2$ and the features if amorphous Si anodes. Based on their results, they showed that the computed average voltage was in alignment with experimental results and validated their model. Onat et al. [118] used an ANN method to represent the atomic interactions of amorphous Li-Si alloys and computed the Li diffusivity to compare their results with the available references. Yoo et al. [119] studied Si crystals as well as nanoclusters with atomic energy mapping modeled in ANN. In a study by Zuo et al. [120] that was conducted to compare the performance of several ML interatomic models, including ANN potential with SF representation as well as GPR potential, it was found out that all the ML potentials illustrated accurate prediction of forces, energies, and thermal properties.

### 3.3.3. Electrolyte Materials

The electrolyte in SSBs is an important component of the battery that should be accurately modeled to possess high ionic conductivity and compatibility with electrodes and mechanical stability. Lacivita et al. [121] devised a set of data-driven methods to determine the N defects in $Li_3PO_4$. Their study focused on the potential energy surface (PES) sampling and used ANN for fast screening. According to the investigation by Li et al. [122] which used ANN models to evaluate Li diffusion in amorphous $Li_3PO_4$, it was shown that including Li diffusion transient structures in the simulations is an essential parameter to reduce the error of the barrier energies. Miwa et al. [123,124] constructed an automatic Bayesian optimization model without including any stochastic assumption. Their study presented that the promotion of Li diffusion in $\beta$-$Li_2B_{12}H_{12}$ is achieved by lattice expansion and predicted the conductivity of Li and activation barriers in Nb-doped LLZO. Deng et al. [79] proposed an electrostatic spectral analysis potential (eSNAP) and simulated the diffusion of Li layers in superionic conductor $\alpha$-$Li_3N$ to provide insight into the concerted ionic motion and grain boundary diffusion. Wang et al. [125] studied the Li diffusion pathways in the interphase and examined the Li ionic conductivities of Li materials on the interfaces of electrolytes using LOTF-MD methods. In a study that was conducted by Fujimura et al. [126], an SVM method was employed to model a diffusion-based model considering the temperature and energy formation and diffusion coefficient to study the ionic conductivity of electrolytes. Due to the importance of fast ionic conductivity and electrochemical stability, Sendek et al. [127] carried out a study that showed the inclusion of Cl-, Br-, and I-based solid ion conductors lead to more efficient stability and ionic conductivity.

### 3.3.4. Comparison of Data-Driven Approaches

Data-driven prognosis and health assessment of SSBs are complex procedures requiring an extensive survey in the system's domain under analysis. System characteristics, data availability, and application constraints are the primary components that need to be taken into consideration before selecting the appropriate method. With that being said, no unique technique can be identified as the most efficient approach as requirements of the users and decision-makers vary from project to project. Thus, it is recommended that the selection of the data-driven method be completely based on the specific system, working environment, and the cost of the assessment.

### 3.4. Hybrid Approaches

Even though physics-based models provide valuable insight about the internal state of SSBs, they require extensive parameter estimation of the components. Furthermore, in some systems, it is not amenable to perform off-line testing of the system to extract measurements of the cells, particularly when the model parametrization is further become challenging by the inherent cell–cell variability. To overcome these bottlenecks, hybrid models are introduced that can be used as an advanced SSB state estimation tool in battery BMS. Hybrid approaches combine physics-based methods with data-driven methods to obtain accurate predictions of the SOH of battery systems. The most commonly used hybrid approaches are series and parallel [128]. Combining a model-based model with available prior knowledge about the process and a data-driven method lead to a series approach. Similarly, in parallel methods, physics-based and data-driven approaches are simultaneously considered from the model. There have been limited research studies that focused on using hybrid methods to estimate the battery state. The bottleneck that is embedded in the vast adoption of hybrid model lies beneath the fact that high-fidelity multiphysical and multiscale models are needed so that it would be possible to train machine-learning models and ultimately create new opportunities for fusing the advantages of both modelling approaches [129]. The studies that are mentioned here are the most prevalent techniques that have been developed in the battery management field. A study by Song et al. [29], implemented a data-driven least-square support vector machine that is combined with a model-based

unscented-particle filter that can be used to increase SOC and SOH of SSBs and LIBs. Lyu et al. [130] used a Thevenin model and a data-driven method to estimate the SOH of batteries. In 2021, Lin et al. [131] proposed a hybrid approach in which a continuous hidden Markov model was combined with kernel density estimation to estimate the SOH of batteries.

The summary of the prognostication methods that are commonly used for SSBs that were discussed above is shown in Table 1.

**Table 1.** Summary of PHM techniques for solid-state batteries.

| Categories | Technique |
|---|---|
| Model-based and Physics-based method | Equivalent circuit model [85]<br>Electrochemical model [61,62,77]<br>Electrochemical-mechanical model [63,64,68,70]<br>Mathematical model [65,72,78] |
| Data-driven method | ANN [104–107,114,115]<br>kNN [108]<br>DDP [111,112]<br>SVM [113]<br>RF [109]<br>PF [87]<br>BM [86,90] |
| Hybrid method | Series [129,130]<br>Parallel [131] |

## 4. Challenges, Perspectives, and Conclusions

PHM-augmented schemes provide an essential characterization framework for evaluating the availability and reliability of SSBs. Even though many of the PHM methods are complex procedures that require extensive characterization and interpretations, they are a crucial component of safety requirements for battery systems. Predictably, the PHM approaches become more commonplace as the desire to transition from fossil fuels to sustainable energy is increasing substantially nowadays. However, the road to achieving this goal is still in need of pavement as the limitations to employing PHM methods for SSBs are high. For instance, most PHM techniques, particularly ML-based data-driven methods, require the modeling to be developed for a specific domain and cannot be generalizable to other instances. This limitation poses a challenge to accurately incorporating the complex processed in SBBs and batteries in general (such as degradation).

Moreover, another hurdle in adopting PHM methods in SSBs is reproducibility. Although PHM methods have gained significant popularity over the recent decades, a lack of sufficient quality measures and material science library for establishing robust models have caused complications in utilizing these methods. A universal solution can be the willingness of researchers to share the model validations and data to accelerate the PHM models' applicability. The lack of reliable data also hurts advances in this area because because most PHM techniques are based on simulation results and not experimental, which can be improved by experimentally validating the simulations results.

This review was carried out to present the current state of the PHM modeling of SSBs with the focus on the application of model-based and ML-based data-driven approaches. Despite vast limitations and challenges that are still ahead, many encouraging PHM-based studies have appeared in the literature. With the ever increasing demand for a safer and more reliable battery module, and ultimately a sustainable energy, PHM frameworks are attracting more attention due to their capability to facilitate battery health assessment. By studying the current state of art of PHM techniques for SSBs, the following conclusion can be made: (1) model-based approaches and data-driven methods are primarily used to estimate the SOC of SSBs that might not be sufficient for accurate state estimation. This is due to the fact that available methods focus on the cell battery rather than the battery pack. This can lead to inaccurate prognostications and health status analysis. In order to improve the robustness of these models, the battery pack should be considered as a whole.

(2) Although RUL prediction is one of the most important components in the BMS of SSBs, very few studies have focused on this field. This can be attributed to the complexity of the model that needs to be established and the lack of accurate data. (3) Hybrid approaches are capable of addressing the shortcomings of model-based and data-driven methods in SSBs, but little information is available in this area.

**Author Contributions:** Conceptualization, H.S.K. and L.L.; methodology, H.S.K., G.Q., X.Y. (Xiaoping Yi), X.L., R.W., Y.G., X.Y. (Xiao Yu) and L.L.; software, H.S.K., G.Q., X.Y. (Xiaoping Yi), X.L., R.W., Y.G., X.Y. (Xiao Yu) and L.L.; validation, H.S.K., G.Q., X.Y. (Xiaoping Yi), X.L., R.W., Y.G., X.Y. (Xiao Yu) and L.L.; formal analysis, H.S.K., G.Q., X.Y. (Xiaoping Yi), X.L., R.W., Y.G., X.Y. (Xiao Yu) and L.L.; investigation, H.S.K., G.Q., X.Y. (Xiaoping Yi), X.L., R.W., Y.G., X.Y. (Xiao Yu) and L.L.; resources, H.S.K., G.Q., X.Y. (Xiaoping Yi), X.L., R.W., Y.G., X.Y. (Xiao Yu) and L.L.; data curation, H.S.K., G.Q., X.Y. (Xiaoping Yi), X.L., R.W., Y.G., X.Y. (Xiao Yu) and L.L.; writing—original draft preparation, H.S.K. and L.L.; writing—review and editing, H.S.K., G.Q., X.Y. (Xiaoping Yi), X.L., R.W., Y.G., X.Y. (Xiao Yu) and L.L.; visualization, H.S.K., G.Q., X.Y. (Xiaoping Yi), X.L., R.W., Y.G., X.Y. (Xiao Yu) and L.L.; supervision, L.L.; project administration, H.S.K., X.L., Y.G. and L.L.; funding acquisition, X.L., Y.G. and L.L. All authors have read and agreed to the published version of the manuscript.

**Funding:** This research was funded by National Science Foundation under Award #1840732, KU RISe Award, KU Research GO awards, KU General Research Funds, National Natural Science Foundation of China (No. 52076012, No. 51676013), key research projects of North Minzu University in 2019 (No. 2019KJ35) and lateral research projects of North Minzu University in 2020 (No. 2020-108, 2020-136).

**Data Availability Statement:** Not applicable.

**Acknowledgments:** L.L. would like to thank the support from the National Science Foundation under Award #1840732, KU RISe Award, KU Research GO awards, and KU General Research Funds. X.L. would like thank the support from the National Natural Science Foundation of China (No. 52076012, No. 51676013). Y.G. would like thank the support from key research projects of North Minzu University in 2019 (No. 2019KJ35) and lateral research projects of North Minzu University in 2020 (No. 2020-108, 2020-136). Any opinions, findings, and conclusions or recommendations that were expressed in this material are those of the author(s) and do not necessarily reflect the views of the funding agencies.

**Conflicts of Interest:** The authors declare no conflict of interest.

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
