# Peer review of "Prognosis and Health Management (PHM) of Solid-State Batteries: Perspectives, Challenges, and Opportunities"

_energies, doi:10.3390/en15186599_

Round 1

Reviewer 1 Report

The authors review the prognosis and health management (PHM) of solid-state Batteries and point out the opportunities and challenges of the current solid-state battery research. The paper is well-written and logical, I recommend that authors address the following minor issues before accepting the manuscript.

In Line 94, the description of the battery model is inappropriate because the model is a floor plane rather than one-dimensional. In my opinion, there is no difference between the concept of an all-solid-state battery and a solid-state battery. I suggest dividing the solid-state battery into the regular solid-state battery, thin film battery, and “3D” battery.

Author Response

We greatly appreciate your time and comments. We have addressed all comments in both reply letter and  revised manuscript, respectively. All changes made in the revised manuscript have been highlighted in yellow

Reviewer 2 Report

Check the attached file.

Author Response

We greatly appreciate your time and comments. We have addressed all comments in both reply letter and revised manuscript, respectively. All changes made in revised manuscript have been highlighted in yellow.

Reviewer 3 Report

Manuscript Number: 1906876

Prognosis and Health Management (PHM) of Solid-State Bat- 2 teries: Perspectives, Challenges, and Opportunities

Reviewer(s)’ General Comments to Authors:

The obtained and discussed data in this paper should be published in the Energies after minor revision.

There are weak points:

1.     The Authors should add some papers in the introduction part, regarding the systematic investigations of thermal runaway of  lithium-ion batteries:

Effects of Minor Mechanical Deformation on the Lifetime and Performance of Commercial 21700 Lithium-Ion Battery, DOI 10.1149/1945-7111/ac79d4

 The results seem to be carried out carefully and thus the data are reliable. The treatment of hypothesis and results is correct and the obtained conclusions are interesting. This paper is suitable for publication in this journal Energies while some improvements in the framework of minor revision as mentioned in the letter above have been made.

Author Response

(The authors gave the same response as above.)

Round 2

Reviewer 2 Report

The authors addressed all my comments in the updated manuscript. The review is original and interesting both from a scientific and technical point of view. Hence, it is recommended for possible publication in Energies.